# A Systematic Review of the Predictive and Diagnostic Uses of Neuroinflammation Biomarkers for Epileptogenesis

**DOI:** 10.3390/ijms25126488

**Published:** 2024-06-12

**Authors:** Maria Jose Aguilar-Castillo, Pablo Cabezudo-García, Guillermina García-Martín, Yolanda Lopez-Moreno, Guillermo Estivill-Torrús, Nicolas Lundahl Ciano-Petersen, Begoña Oliver-Martos, Manuel Narváez-Pelaez, Pedro Jesús Serrano-Castro

**Affiliations:** 1Servicio de Análisis Clínicos, Hospital Regional Universitario de Málaga, 29010 Málaga, Spain; maijo.aguilarcastillo@gmail.com; 2Instituto de Investigación Biomédica de Málaga y Plataforma de Nanomedicina-IBIMA Plataforma BIONAND, 29590 Málaga, Spain; pablocabezudo@gmail.com (P.C.-G.); guillerminagmartin@gmail.com (G.G.-M.); yolandalopm@gmail.com (Y.L.-M.); estivill.guillermo@gmail.com (G.E.-T.); nicolundahl@yahoo.es (N.L.C.-P.); begoliver@gmail.com (B.O.-M.); 3Servicio de Neurología, Hospital Regional Universitario de Málaga, 29010 Málaga, Spain; 4Alianza Andalucía Neuro-RECA—Roche en Neurología Médica de Precisión, 29010 Málaga, Spain; 5Hospitales Vithas Málaga y Xanit Internacional, 29016 Málaga, Spain; 6Departamento de Fisiologia Animal, Biologìa Celular y Genética, Universidad de Málaga, 29010 Málaga, Spain; 7Departamento de Fisiología, Universidad de Málaga, 29010 Málaga, Spain; 8Departamento de Medicina y Dermatología, Universidad de Málaga, 29010 Málaga, Spain

**Keywords:** epileptogenesis, neuroinflammation, biomarkers, HMGB1, TNF-a, TLR-4, sTNFr2, CCL2, IL-33, drug resistant epilepsy

## Abstract

A central role for neuroinflammation in epileptogenesis has recently been suggested by several investigations. This systematic review explores the role of inflammatory mediators in epileptogenesis, its association with seizure severity, and its correlation with drug-resistant epilepsy (DRE). The study analysed articles published in JCR journals from 2019 to 2024. The search strategy comprised the MESH, free terms of “Neuroinflammation”, and selective searches for the following single biomarkers that had previously been selected from the relevant literature: “High mobility group box 1/HMGB1”, “Toll-Like-Receptor 4/TLR-4”, “Interleukin-1/IL-1”, “Interleukin-6/IL-6”, “Transforming growth factor beta/TGF-β”, and “Tumour necrosis factor-alpha/TNF-α”. These queries were all combined with the MESH terms “Epileptogenesis” and “Epilepsy”. We found 243 articles related to epileptogenesis and neuroinflammation, with 356 articles from selective searches by biomarker type. After eliminating duplicates, 324 articles were evaluated, with 272 excluded and 55 evaluated by the authors. A total of 21 articles were included in the qualitative evaluation, including 18 case–control studies, 2 case series, and 1 prospective study. As conclusion, this systematic review provides acceptable support for five biomarkers, including TNF-α and some of its soluble receptors (sTNFr2), HMGB1, TLR-4, CCL2 and IL-33. Certain receptors, cytokines, and chemokines are examples of neuroinflammation-related biomarkers that may be crucial for the early diagnosis of refractory epilepsy or may be connected to the control of epileptic seizures. Their value will be better defined by future studies.

## 1. Introduction

Neuroinflammation refers to the inflammatory response of the central nervous system (CNS) to deviations from homeostasis that cannot be reversed by homeostatic mechanisms alone [1]. Today, it is known that neuroinflammation is one of the pathophysiological processes that is transversally involved in multiple pathologies of the CNS [2]. So, its role has been shown to be relevant in the pathogenesis of diseases such as multiple sclerosis [3], as the most paradigmatic example, but it is present in a wide variety of other diseases, with special attention to neurodegenerative diseases [4,5].

Resident innate immune cells (microglia and astrocytes), cytokines, and their receptors and chemokines are involved in this CNS inflammatory response. Understanding the processes that occur between the immune system and CNS is crucial, especially in the age of personalised medicine, since many of the elements identified can become diagnostic or prognostic biomarkers of the disease or even become therapeutic targets [6,7,8]. 

Epileptogenesis is the process by which a normal neural network becomes hyperexcitable and capable of causing seizures of epilepsy on its own [9]. In recent years, several studies have addressed the role that neuroinflammation may play in epileptogenesis [10,11,12]. Although the molecular mechanisms underlying these pathophysiological processes are not yet fully understood, it has been speculated that inflammatory mediators may cause abnormal angiogenesis and impairment of permeability of the blood–brain barrier (BBB), a circumstance that is closely related to epileptogenesis [12,13]. On the other hand, the unregulated focal or systemic inflammatory processes themselves lead to the formation of aberrant neuronal connections and hyperexcitable neural networks, as well as an altered response to neurotransmitters, thus participating in the process of epileptogenesis. For both reasons, over the past two decades, there has been growing evidence of both clinical and basic studies providing strong support for the conclusion that neuroinflammation is involved in epileptogenesis [14,15,16,17,18,19,20,21]. Finally, the role of cytokines as potential pro-inflammatory mediators in the neuropathology of epilepsy may also contribute to elucidating the process of epileptogenesis [22,23,24,25]. Thus, the overexpression of some of these inflammatory mediators in the hippocampus and neocortex of patients with epilepsy indicates the activation of multiple pro- and anti-epileptogenic immune pathways.

The cross-functional nature of all of these neuroinflammatory processes across various neurologic diseases may help to explain the high incidence of epileptic seizures in patients with neurological pathologies as diverse as multiple sclerosis [26] or neurodegenerative disease [27]. Figure 1 clarifies the idea that neuroinflammatory mechanisms could be the mediators of epileptogenesis. 

We know that between 25 and 30% of epileptic patients are resistant to available anti-seizure medications (ASMs) [28], and some studies have shown that levels of inflammatory mediators may be elevated in these patients [7,16,29]. These data could open the door to the use of these molecules as predictive biomarkers of DRE and for the development of new treatments for these patients [30]. 

The purpose of this systematic review is to know the state of art in the comprehension of the role of inflammatory mediators in epileptogenesis, to know if elevated levels of inflammatory mediators in serum and CSF can be associated with seizure severity and recurrence, and to know which of them can be most correlated with DRE.

## 2. Methods

### 2.1. Search Strategy and Databases 

A systematic search was carried out using the following inclusion criteria:Articles published in journals indexed in JCR in the last 5 years (January 2019 to February 2024) using the MEDLINE database.Articles that included retrospective, prospective, case–control, or cross-sectional studies.Searches were carried out by combining the following MESH and free terms: “Epileptogenesis” and “Neuroinflammation”.To obtain a more selective search for certain molecules of special interest, we included searches aimed at biomarkers that had previously been reported in narrative reviews on the topic [12,22,31,32,33,34]. Figure 1 shows the specific role of these biomarkers in the hypothesis of the epileptogenesis of neuroinflammatory origin. Specifically, the searches included in this additional search were:“High mobility group box 1/HMGB1” AND “Epilepsy”“Toll-Like-Receptor 4/TLR-4” AND “Epilepsy”“Interleukin-1/IL-1” AND “Epilepsy”“Interleukin-6/IL-6” AND “Epilepsy”“Transforming growth factor beta/TGF-β” AND “Epilepsy”“Tumour necrosis factor-alpha/TNF-α” AND “Epilepsy”

We used the more general term “Epilepsy” instead of “Epileptogenesis” to increase the sensitivity of the search.

5.Articles published in English and/or Spanish.

The review protocol followed the declaration of Preferred Reporting Items for Systematic Reviews and Meta-Analyses (PRISMA) [35]. 

### 2.2. Exclusion Criteria

Duplicate articles, editorials, letters to the editor, or narrative reviews.Articles that included basic research studies on tissues or animal models (the “human” filter in MEDLINE searches was used).Articles focused on acute symptomatic seizures due to infectious, traumatic, vascular or oncological processes.Articles focused only on the treatment of epilepsy.Articles that analysed biomarkers of epilepsy in the setting of general brain damage.Articles focused on autoimmune epilepsy or epilepsy in neurodegenerative diseases.

### 2.3. Study Selection

Each of the selected articles was evaluated by reading the title, abstract, and their keywords by two different reviewers (MJA and PSC). We did not use any particular programme to review the literature. After carefully examining the title and abstract, the inclusion and exclusion criteria outlined in the preceding section were strictly adhered to. In the case of discrepancy between the two reviewers, the aid of a third reviewer (PCG) was needed. The selected articles were read in their entirety to evaluate the degree of scientific evidence. The included studies were assessed based on the criteria discussed in the Standards for Reporting of Diagnostic Accuracy (STARD) checklist [36]. 

### 2.4. Data Extraction

The required data from the inputs were extracted by one of the researchers (MJA) into Table 1. The level of evidence was assessed using the Scottish Intercollegiate Guidelines Network (SIGN) grading system [37].

### 2.5. Flowchart 2020 PRISMA



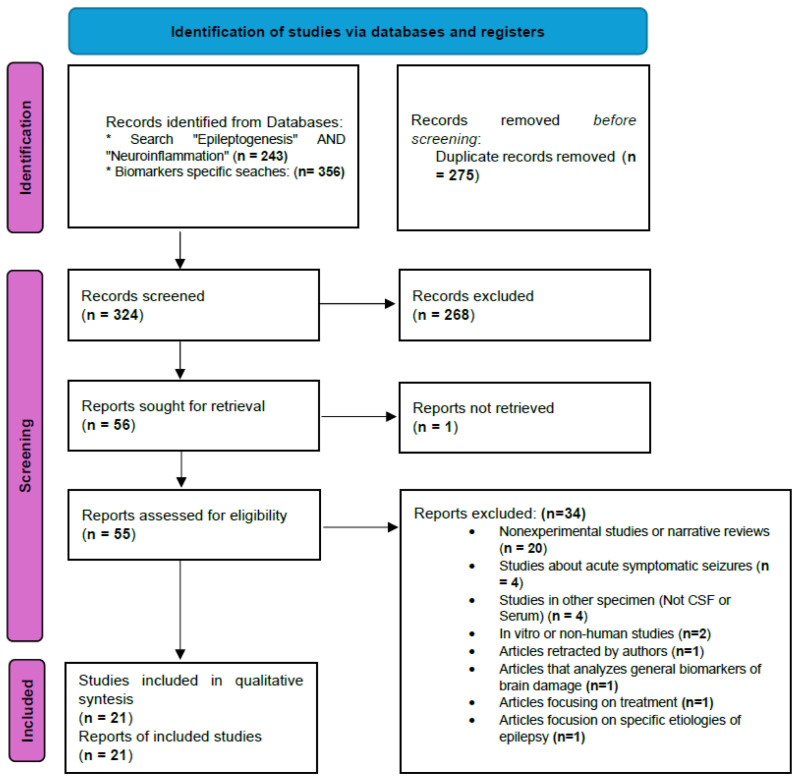



## 3. Results and Discussion

### 3.1. Overall Results of the Literature Search

A total of 243 articles resulted from the search “epileptogenesis” AND “neuroinflammation” and 356 from selective searches by biomarker type. Specifically, the numbers of results by type of biomarker were:HMGB1: 31 resultsTLR-4: 38 resultsIL-1b: 69 resultsIL-6: 98 resultsTGF-β: 29 resultsTNF-a: 91 results

After eliminating duplicates, 324 articles were evaluated by reading their titles and abstracts and applying the defined inclusion and exclusion criteria. A total of 272 articles were excluded and another 55 were evaluated by the authors through reading the full text. Finally, 21 articles were included in the qualitative evaluation. See the PRISMA 2020 Flowchart. By the type of article, our selection included 18 case–control studies, 2 case series, and 1 prospective population-based study.

### 3.2. Results by Specific Biomarkers

#### 3.2.1. Interleukin 1β (IL-1β)

Our review found studies with conflicting results. The study with the highest level of evidence was a prospective study by Wang et al. [48] that did not show elevated levels of this biomarker in people with epilepsy. Other case–control studies also reported similar results [38,56]. Choi et al. [40], conversely, in a retrospective case–control study, found an elevation of serum and CSF IL-1β levels in children who had suffered epileptic seizures in the previous 48 h. Kamaşak et al. [32] studied 28 children with DRE of at least three years of evolution, demonstrating significant differences in IL-1β levels. Based on these results, it has been proposed as a biomarker of drug-resistance. Other studies, however, reported decreased levels of this cytokine in DRE [41]. Our conclusion is that there is no conclusive evidence in this regard, although it is possible that IL-1β could be a valid marker for the detection of recent seizures. 

#### 3.2.2. Interleukin 6 (IL-6)

We found five case–control studies [44,48,49,56,57], of which only one of them [44] found a decrease in serum levels in patients with epilepsy compared to healthy controls. For this reason, the role of this IL as a biomarker is not fully defined. 

#### 3.2.3. Interleukin 17 (IL-17)

A prospective study on patients with TLE [48] and three other case–control studies included in this systematic review [44,49,56] determined this IL, but only the study by Alvim et al. [44] showed a decrease in its serum levels of patients with epilepsy compared to healthy controls. We consider that there is insufficient evidence to defend its role as a useful biomarker in clinical practice. 

#### 3.2.4. Other Interleukins

A significant additional number of ILs have eventually been studied in case–control studies. We must highlight the study by Alvim et al. [44], which shows a decrease in serum levels of IL-2, IL-4, IL-6, or IL-10 in patients with epilepsy versus healthy controls. In the study by Ethemoglu et al. [46], IL-33 levels were found to be higher in all patients with epilepsy.

#### 3.2.5. TNF-α

In total, 11 publications that were part of our systematic review contained this molecule. Most of them did not demonstrate any appreciable distinction between epilepsy sufferers and healthy controls. However, three of them—the TLE study by Wang et al. [48], the DRE study by Sokolova et al. [50], and the chronic epilepsy study by Kothur et al. [38]—did indicate an increase in these levels in these patients. According to this last study, there was an increase in CSF TNF-α levels during the acute stage of status epilepticus or FIRE-type encephalitis and a decrease during the chronic phases. We believe that there is some bibliographic support to consider that this biomarker may be related to DRE.

#### 3.2.6. Transforming Growth Factor Beta (TGF-β)

Several investigations have been carried out to establish a connection between this molecule and DRE. In a case–control study, Yu et al. [58] demonstrated that individuals with DRE had higher TGF-β levels in their CSF than patients with controlled seizures. Since this article was published in 2014, it was not incorporated into the systematic review. There are no other studies on this biomarker included in the review, so we cannot prove its validity in patients with epilepsy.

#### 3.2.7. Toll like Receptor 4 (TLR-4)

Three of the case–control studies included in this review demonstrated the association of high levels of this biomarker with DRE [32,39,45], as well as its correlation with the number of seizures [39]. So, it can be proposed as a biomarker not only of refractoriness but also of the control of epilepsy.

#### 3.2.8. HMGB1

In our review, we found up to six case–control studies that have shown that HMGB1 levels are increased in the blood of patients with epilepsy compared to healthy individuals [39,42,45,47,51,53]. One of these studies analysed concomitantly the levels of this molecule in CSF and serum and found elevation of its levels also in CSF, but without correlation with serum values [51]. 

A further question some research addressed is whether the increase only occurs immediately following a seizure or if it lasts for hours afterwards. Blood samples were taken by Nass et al. [53] at the start of the investigation, as well as 2, 6, and 24 h following a generalised tonic-clonic seizure. The findings verified that HMGB1 levels were raised right away and continued to be raised for six hours.

Some studies attempted to characterise this biomarker in DRE. So, Yue et al. [39] found that serum HMGB1 levels were higher in patients with DRE compared to drug-responsive epilepsy. In addition, there was a positive correlation between HMGB1 expression and seizure frequency.

Walker et al. [42] discovered that DRE patients’ serum had higher levels of HMGB1 than that of healthy controls and patients with drug-responsive epilepsy who had not experienced a seizure in at least six months. Still, no meaningful correlation was found with epilepsy subtype or with the length or frequency of seizures during the preceding month. Kan et al. [45] also found a relationship between seizure frequency and duration.

Conflicting results were observed in one study that was part of our review. Panina et al. [47] discovered that, in contrast to a control group, patients with both controlled and refractory temporal lobe epilepsy (TLE) had decreased serum HMGB1 levels. The authors contended that because their sample of patients had a wide range of epilepsy types and durations, their results were inconsistent with those of other studies.

These studies suggest that HMGB1 can be used as a biomarker of DRE with a greater number of published studies supporting its use than other inflammatory markers included in this review.

#### 3.2.9. Chemokines

In several studies of our review [38,49,54,55,57], levels of some chemokines were found to be increased. The one that was identified in the largest number of studies was CCL2, a potent chemoattractant protein for monocytes, and hence is alternatively referred to as monocyte chemoattractant protein-1 (MCP-1). Elevated levels of CCL2 expression in an inflamed brain were associated with activation and recruitment of macrophages/microglia to the injury sites [59]. Česká et al. [57] compared cytokine levels in children with DRE with healthy control and control patients in both CSF and blood plasma, finding elevated levels of CCL2. For this reason, we conclude that CCL2 can be proposed as a possible biomarker of DRE.

#### 3.2.10. Soluble TNF-α Receptors

According to the study by Alvim et al. [44], individuals with epilepsy had greater plasma levels of sTNFr2 than controls, but lower levels of TNFα and sTNFr1. Accordingly, sTNFr2 plasma levels are a promising indicator of epilepsy activity.

### 3.3. Discussion

Neuroinflammation is a transversal process that underlies a variety of neurological disorders, including those with immunological etiology (multiple sclerosis [60]), but also neurodegenerative [8] (Alzheimer’s disease, Parkinson’s disease, etc.) and even neuropsychiatric ones, such as schizophrenia or chronic depression [61]. Epilepsy is more common in all of these disorders than in the general population [62]. This epidemiological data may be partially explained by a shared pathophysiology substrate. 

Finding neuroinflammatory biomarkers associated with epileptogenesis in the literature was the original goal of our review. 

Our preliminary analysis showed that no very high-level evidence-based research has been conducted. Out of the 21 selected articles, 18 were case–control studies, 2 were uncontrolled case series studies [53,55], while one was a cohort study [48], and though two of them analyses CSF at the same time [51,57], the great majority concentrated on serum analysis and another only analysed CSF [38]. In addition, there was significant variation across the patient cohorts utilised in the various studies regarding the type of epilepsy included, the patient inclusion criteria, and the total number of patients employed. Because of this, there was a lot of variation in the findings, which makes comparing these studies challenging. 

Despite these limitations, this systematic review provides acceptable support for five biomarkers: TNF-α and some of its soluble receptors (sTNFr2), HMGB1 and its TLR-4, CCL2 and IL-33 (Table 2).

Certain autoimmune illnesses have been linked to increased amounts of the nuclear non-histone protein HMGB-1. Its two main receptors, TLR-4 and RAGE, mediate a biological process that stimulates inflammation [63] by starting the IL-1R/TLR-4 cascade, which produces some types of cytokines and raises the hyperexcitability of neurons [64] (See Figure 1). Furthermore, because it stimulates phosphorylation of the N-methyl-D-aspartate receptor (NMDAR) and raises the Ca2+ channel’s permeability, the HMGB1–TLR-4 axis is essential for neuronal hyperexcitability. The breakdown of the blood–brain barrier (BBB) can also result from activation of the HMGB1–TLR-4 axis, allowing the permeation of activated lymphocytes, antibodies, inflammatory cytokines, or albumin from peripheral blood to the CNS, aggravating inflammation. Elevated HMGB1 levels have been associated with other CNS disorders [65,66,67,68]. The results of our review position this biomarker as the most supported by recent studies, especially in the characterisation of DRE [39,59] and as a biomarker of recent epileptic activity [53]. 

TLR-4 is an innate immune system receptor of HMGB1 y that has been proposed as a promoter of epileptogenesis [69] (See Figure 1). The results of our systematic review allow us to propose this molecule as a biomarker of DRE. 

Another molecule of interest in our work is the pro-inflammatory cytokine TNF-α. This molecule is secreted by activated astrocytes and microglia; it raises microglial glutamate levels and promotes Ca2+ entry [25] (See Figure 1). Our systematic review found several studies that correlated this serum biomarker with DRE or chronic epilepsy [38,48,50].

Perhaps of greater interest is the identification of some of TNF-α receptors as biomarkers. There are two types of TNF-α receptors: membrane-bound (TNFr1 and TNFr2) and soluble (sTNFr1 and sTNFr2). Some studies suggest that sTNFr1 and sTNFr2 better reflect TNFα activity because they are more stable than TNFα. The results of our review corroborate that sTNFr2 could be a promising biomarker related to epileptic activity [44]. 

Astrocytes, microglia, and endothelial cells secrete a family of proteins known as chemokines. By controlling neurotransmitter-releasing, voltage-dependent, or G-protein-dependent channels, they modulate neuronal excitability and are crucial to the entry of immune cells into the brain via certain G-protein-coupled receptors (GPCRs) [70,71]. This accumulating evidence suggests that chemokines and downstream signalling pathways mediate the interaction between neuroinflammation and epileptogenesis. Levels of some chemokines, such as CCL2, CCL3, lCCL4, Fractalkine (CX3CL1), and CXCL13, and the corresponding chemokine receptors CCR2, CCR5, C-X-C receptor 4 (CXCR4), and CXCR5 have been found to be elevated in the hippocampus in animal models of epilepsy [72]. Elevated levels of CCL2 expression in an inflamed brain were associated with activation and recruitment of macrophages/microglia to the injury site [59]. The results of our systematic review only allow us to propose CCL2 as a biomarker of epilepsy [49,55,57], a potent chemoattractant protein for monocytes, and hence is alternatively referred to as monocyte chemoattractant protein-1 (MCP-1). 

Even if some of the other molecules under examination were preceded by encouraging experimental trials, there does not seem to be enough information about them at this moment to consider them as prospective biomarkers for persistent epilepsies. 

So, Il-1b is a cytokine produced by the resident cells of central innate immunity. The binding of IL-1β to its receptor (IL-1R) activates nuclear factor κB (NF-κB) and three mitogen-activated protein kinase (MAPK) signalling pathways, all of which are involved in the production of other cytokines and in the upregulation of genes related to inflammation and generation of reactive oxygen species (See Figure 1). These signalling pathways lead to the activation of TLR-4 [34,73]. On the other hand, IL-1β influences the entry of calcium through the NMDAR, reduces glutamate uptake by astrocytes, and increases glutamate release by glial cells [74]. Both TLR-4 and IL-1β are expressed at low levels within the brain under basal conditions but can increase rapidly during acute pathological conditions. Thus, elevated levels of IL-1β have been found after febrile seizures [75] and its role in epileptogenesis has been postulated [76]. However, our systematic review has found contradictory results that prevent us from including this molecule as a consolidated biomarker in epileptogenesis right now. Further studies will be needed to decide the true role of this biomarker.

IL-6 plays a fundamental role in regulating the inflammatory response and activating adaptive immunity [77]. Astrocytes and neurons synthesise this substance, which can be released by perivascular and brain endothelial cells in response to inflammatory and infectious stimuli. Some older studies not included in our review suggested that IL-6 could be elevated in CSF and plasma immediately after tonic-clonic seizures [78,79] or in patients with focal epilepsies [80], but our systematic review failed to corroborate the results of these studies.

Of the rest of the interleukins, only the study by Ethemoglou et al. [46] showed that IL-33 levels were higher in all patients with epilepsy. IL-33, a novel member of the cytokine family associated with IL-1, is widely expressed from a wide range of cell types and tissues, including astrocytes, neurons, microglia, and oligodendrocytes.

Finally, TGF-β is involved in cell proliferation, growth, and differentiation. Its role in neuroinflammation, however, is still unclear. The TGF-β type I receptor is more highly expressed in the cytoplasm of astrocytes, according to research on neocortical temporal lobe tissue from 30 epileptic patients [81]. It has also been suggested that blocking this receptor in vivo reduces the likelihood of epileptogenesis [82]. Our systematic review does not allow us to propose it as a biomarker of epilepsy in view of the current evidence. 

The role of inflammatory markers as biomarkers of epilepsy has already been discussed in previous reviews, such as by Zhang et al. [69]. The connection between two signalling pathways, the HMGB1/TLR4 and IL-1β/IL-1R1 pathways, and epilepsy is the main topic of this review. The study also looks at the mediators’ roles and their pathways of sequestration and acknowledges the need for more research on the possibility of cell-to-cell interference in neuroinflammation, including activated microglia, astrocytes, and neurons, as well as the function of pro-inflammatory substances like chemokines, cytokines, bioactive lipids, and growth factors.

#### 3.3.1. Potential Clinical Practice Applications

In general, the goal of applying personalised medicine principles to neurological diseases is to improve the identification of conditions that significantly affect quality of life, hence increasing the window of opportunity for therapeutic intervention. Neurodegenerative illnesses are one example of this [83,84]. Alzheimer’s disease biomarker research has already been applied in clinical settings and is directly affecting the treatment of these patients [84]. In the field of epilepsy, the fundamental challenge would be the early identification of DRE. This is a type of epilepsy that affects approximately one-third of patients and that may, in certain cases, have a non-pharmacological solution, such as epilepsy surgery or certain neuromodulation techniques [85]. At present, there is a significant delay in the detection of refractoriness and, therefore, there is a significant delay in applying effective treatments. One of the main consequences of our study is the possible consideration of the TNF-a/sTNFr2 and HMGB1/TLR-4 axes as early biomarkers of DRE. Its possible use in the future should be established by studies specifically designed for this purpose.

#### 3.3.2. Future Lines of Development and Research

The identification of biomarkers involved in epileptogenesis may open the door to the development of specific treatments capable of modifying the natural history of the disease. Examples of using biomarkers as treatment targets in the context of personalised medicine for highly specific forms of epilepsy have been published in the past [86,87].

There is a previous systematic review on this topic [88]. Although the evidence found by these authors was also scarce, they concluded that the use of anti-IL-1, anti-IL-6, and anti-CD20 agents in patients with DRE and refractory status epilepticus has shown promising results and a good safety profile. 

Some articles have speculated on the possible effect of anti-TNF-α drugs. The experience was limited to published case series in the treatment of Rasmussen’s encephalitis with adalimumab [89,90]. Indeed, there is an ongoing clinical trial trying to evaluate the benefit of adalimumab in patients with Rasmussen’s encephalitis (ClinicalTrials.gov Identifier: NCT04003922). 

Glycyrrhizin, an HMGB1 inhibitor medication, has shown neuroprotective and antiepileptic benefits in many animal models of epilepsy [91,92]. Nevertheless, this medication has not been evaluated in epilepsy clinical studies. 

Since it was not the purpose of our systematic review, it does not add to the body of knowledge on this subject. However, several studies showed promise for providing viable pharmaceutical substitutes for DRE because of its anti-neuroinflammatory characteristics. Among these include selective TLR pathway blockade strategies (perhaps the most promising is resveratrol [93]), Janus kinase/signal transducer and activator of transcription (JAK-STAT) inhibitors drugs [94], NF-kb inhibitor drugs [95], MAPK pathway inhibitor drugs [96], or Purinergic signalling modulation drugs [97].

## 4. Conclusions

Despite the lack of strong scientific backing in the studies that are currently available, they do enable us to identify a number of neuroinflammation-related biomarkers that could be important in the early detection of DRE or could be connected to the level of control experienced by epilepsy patients. Some DAMP molecules like HMGB1 and its membrane receptors (TLR-4) are among them, as are cytokines like TNF-a and some of its soluble receptors, like sTNFr2 or IL-33, as well as chemokines like CCL2. These biomarkers may occasionally be suggested as potential treatment targets for particular forms of epilepsy. Future research will enable us to better characterise their actual value.

## Figures and Tables

**Figure 1 ijms-25-06488-f001:**
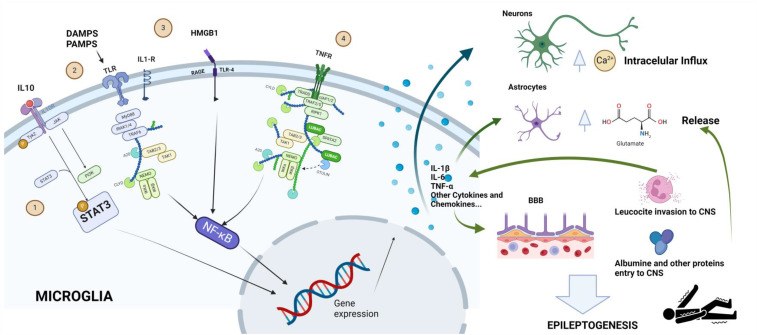
Some of the most important intracellular molecular pathways leading to epileptogenesis are shown. 1.—JAK-STAT pathway: proinflammatory cytokines (such as IL-10) set this pathway in motion. The phosphorylated STAT3 protein can alter gene expression to induce secretion of proinflammatory cytokines (IL-1β, IL-6, or TNF-α). 2.—The TLR pathway can be activated by DAMPS or PAMPS. 3.—HMGB1 is produced by cell damage and activates the pathway through its interaction on RAGE-TLR4. 4.—TNFR can be activated by the release of TNF-α. The pathways activated by TLR, RAGE-TLR4, and TNFR converge in the activation of NF-κβ, which also modifies gene expression to induce the secretion of proinflammatory cytokines. The release of these pro-inflammatory molecules acts on the BBB, causing leukocyte penetration, on astrocytes, causing the release of glutamate, the primary excitatory neurotransmitter, and on neurons, increasing their excitability through an increase in calcium influx. Proteins like albumin also feed back into the release of pro-inflammatory cytokines and glutamate. Modified from Aguilar-Castillo et al. [12]. Created with www.Biorender.com (accessed on 2 June 2024).

**Table 1 ijms-25-06488-t001:** Summary of the studies included in the systematic review for qualitative analysis. Only significant results are shown. The distinction between risk of bias in studies classified as level 2 (2− vs. 2+) was made based on criteria related to sample selection, total number of patients, number of healthy controls, and the clarity of description of procedures.

Reference	Biomarkers Studied	Sample Type	Type of Study	Results	N	Level of Evidence
Kothur et al. [38]	IL-1ra, GM-CSF, IL-1β, TNF-α, IL-2,IL-4, IL-6, IL-8, IL-10, IL-13, IL-17A, IFN-γ, CCL2/MCP-1, CCL5/RANTES, CXCL1/GRO, CXCL10/IP-10,CCL3/MIP-1a, CCL4/MIP-1b, IL-12 (p40), IL-12 (p70), IFN-α, G-CSF, CCL11/eotaxin. IL-21, IL-23, CXCL13/BCA-1, CCL17/TARC, CCL21/6Ckine, CXCL12/SDF-1. CXCL9/MIG, CXCL11/I-TAC, and CCL19/MIP-3b	CSF	Case–control	TNF-α and CCL19 levels were mildly elevated in chronic epilepsy.	6 Patients with FIRES-related disorders vs. 8 Febrile status epilepticus vs. 8 Afebrile status epilepticus vs. 21 patients with chronic epilepsy	2+
Yue et al. [39]	HMGB1 y TLR4	Serum	Case–control	HMGB1 y TLR4 levels were elevated in chronic epilepsy	72 epilepsy patients with epilepsy vs. 43 healthy controls	2+
Choi et al. [40]	α-synuclein, IFN-β, IFN-γ, IL-1β, IL-6, IL-10, and TNF-α	Serum	Case–control	IL-1β levels were correlated only with the numbers of ASMs used, suggesting DRE.	115 children with epilepsy vs. 10 demyelinating disorders of the CNS vs. 146 healthy controls.	2+
Saengow et al. [41]	gamma (IFN-c), IL-1β, and TNF-a	Serum	Case–control	IL-1β levels weredecreased in patients with DRE. IFN-c levels were increased in patients with DRE.	65 patients with DRE vs. 6 healthy controls	2-
Walker et al. [42]	HMGB1	Serum	Case–control	HMGB1 levels were elevated in DRE	65 patients with DRE vs. 74 healthy controls	2+
Kamaşak et al. [32]	HMGB-1, TLR-4, IL-1R1, TNF-a, and IL-1β	Serum	Case–control	HMGB-1, TLR4, TNF-α, and IL-1β levels were increased in the severe epilepsy group (similar to DRE)	28 children with DRE vs. 29 children with controlled epilepsy vs. 27 healthy controls	2+
Aline et al. [43]	TNF-a, Caspase, and Lipid factors	Serum	Case–control	No significant differences	43 epileptic patients vs. 41 healthy controls	2+
Alvim et al. [44]	IL-1, IL-2, IL-4, IL-6, IL-10, IL-17, IFNγ, TNF-α, soluble TNF receptor 1 (sTNFr1), sTNFr2, BDNF, neurotrophic factor 3 (NT3), NT4/5, ciliary neurotrophic factor (CNTF), nerve growth factor (NGF), and glial cell line-derived neurotrophic factor (GDNF).	Serum	Case–control	BDNF, NT3, NGF, and sTNFr2 levels were elevated in patients with epilepsy IL-2, IL-4, IL-6, IL-10, IL-17, IFNγ, TNFα, and CNTF levels were decreased in patients with epilepsy	446 patients with epilepsy vs. 166 healthy controls.	2+
Minchen et al. [45]	HMGB1 and TLR4	Serum	Case–control	HMGB1 and TLR4 levels were higher in patients with epilepsy and with DRE	51 patients with DRE vs. 54 patients with responsive epilepsy vs. 100 healthy controls	2+
Ethemoglu et al. [46]	IL-33	Serum	Case–control	IL-33 levels were elevated in patients with DRE and responsive epilepsy	21 patients with DRE vs. 39 patients with responsive epilepsy vs. 35 healthy controls	2+
Panina et al. [47]	BDNF, TNF-*a,* HMGB1, and NTRK2	Serum	Case–control	BDNF, TNF-*a*, and HMGB1 levels were decreased in patients with TLE.	49 patients with drug-resistant TLE vs. 117 patients with responsive TLE vs. 203 healthy controls.	2+
Wang et al. [48]	IL-1β, IL-5, IL-6, IL-8, IL-17, IFN-γ, and TNF-α	Serum	Prospective, population-based study	TNF-α levels were elevated in patients with medial TLE (mTLE)-with hippocampal sclerosis (HS) vs. rest of patients TNF-α levels were elevated in patients in the mTLE without HS vs. healthy control group.	30 patients with mTLE with HS vs. 41 patients with mTLE without HS vs. 20 healthy controls	2++
Milano et al. [49]	IL-6, TNF-α, IL-33, IL-8, CCL2, IL-13, IL-1β, IFN-γ, IL-1Ra, CCL3, IL-4, CCL4, IL-5, IL-1α, IL-17 A, IL-18, IL-33r, IL-1RII, and IL-1RI	Serum	Case–control	CCL2, CCL3, and IL-8 levels were elevated in patients with mTLE	25 patients with drug-resistant mTLE vs. 21 patients with drug-responsive mTLE vs. 25 healthy controls	2+
Sokolova et al. [50]	IL-1RA, interferon IFN-, IL-10 IL-2, IL-8, IL-7, TNF-α, IL-4, and sCD40L	Serum	Case–control	IL-2 and IL-8 levels were decreased in DRE patients. TNF-α, IL-4, and sCD40L levels were increased in DRE	6 DRE patients vs. 5 healthy controls.	2−
Wang et al. [51]	HMGB1	CSF and Serum	Case–control	CSF HMGB1 levels were elevated in patients with DRE. CSF HMGB1 levels were elevated in patients with symptomatic etiology. CSF HMGB1 at one-year follow-up in patients with active epilepsy. CSF HMGB1 levels were positively associated with seizure frequency.	27 patients with DRE vs. 56 patients with newly diagnosed epilepsy vs. 22 controls with other non-inflammatory neurological disorders	2−
Mochol et al. [52]	IL-18; Interleukine 18 binding protenin (IL-18BP)	Serum	Case–control	IL-18 and IL-18BP levels were increased in patients with epilepsy	119 patients with epilepsy vs. 80 healthy controls	2+
Nass et al. [53]	c-reactive protein (CRP), HMGB1, S100, RAGE, ICAM1, and MMP9	Serum	Case Series	HMGB1and S100 levels were increased in postictal period	28 patients with epilepsy with generalised seizures.	3
Gakharia et al. [54]	CCL2, CCL4, CCL11, and PGE2	Serum	Case–control	CCL11 and PGE2 levels were increased in patients with DRE.	20 patients with DRE vs. 20 patients with responsive epilepsy vs. 16 healthy controls.	2+
Bronisz et al. [55]	MMP-9, MMP-2, CCL-2, S100B, TIMP-1, TIMP-2, ICAM-1, TSP-2, and P-selectin	Serum	Case series	MMP-2, MMP-9, and CCL-2 levels were related to seizure number in 1, 3, 6, and 12 months of observation.	49 patients with epilepsy.	3
Gledhill et al. [56]	CRP, calbindin, cytokeratin-8, eotaxin, eotaxin-2, eotaxin-3, granulocyte-macrophage colony-stimulating factor, ICAM-1, IFN–γ, IL-1β, IL-1α, IL-2, IL-4, IL-5, IL-6, IL-7, IL-8, IL-10, IL-12/IL-23 p40, IL-12 p70, IL-13, IL-15, IL-16, IL-17, IFN-γ-inducible protein 10, macrophage colony-stimulating factor (M-CSF), monocyte chemoattractant protein (MCP)–1, MCP-2, MCP-4, macrophage-derived chemokine, macrophage migration inhibitory factor, macrophage inflammatory protein (MIP)–1β, MIP-1α, MIP-5, matrix metalloproteinase (MMP)–1, MMP-3, MMP-9, Nectin-4, Osteoactivin, osteonectin, P-cadherin, serum amyloid protein A, stem cell factor (SCF), thymus and activation regulated chemokine, TNF–α, TNF-β, TNF–r1, TNF–r2 (R2), TNF–related apoptosis-inducing ligand (TRAIL), vascular cell adhesion molecule 1, and vascular endothelial growth factor A	Serum	Case–control	TRAIL, ICAM-1, MCP-2, and TNF-r1 levels were increased in patients with epilepsy within 24 h after seizure	137 patients with epilepsy vs. 29 healthy controls.	2+
Česká et al. [57]	IL-6, IL-8, IL-10, IL-18, CXCL10/IP-10, CCL2/MCP-1, BLC, TNF-α, C-X3-X, and fractalquine (CXC3CL1)	CSF and Serum	Case–control	CSF CCL2/MCP-1 levels were increased in patients with DRE Serum Fractalkine/CXC3CL1 levels were elevated in patients with DRE	26 patients with epilepsy (22 DRE, 4 non-DRE) vs. 9 healthy controls.	2−

**Table 2 ijms-25-06488-t002:** Summary of biomarkers that have shown some evidence in the systematic review without contradictory results.

Biomarkers	Number of Studies with Positive Results	Sample	Quality of the Evidence	Conclusions of the Studies
HMGB1	6 case–control studies	Serum and CSF	2+	Possible biomarker of DRE. Possible biomarker of seizure frequency. Temporal relationship with generalised tonic-clonic seizures.
TNF-α	2 case–control studies and 1 prospective population-based study	Serum and CSF	2+	Possible biomarker of DRE.
TLR-4	3 case–control studies	Serum	2+	Possible biomarker of DRE. Possible biomarker of seizure frequency.
rTNFr2	1 case–control study	Serum	2−	Possible biomarker of seizure frequency.
CCL2/MCP-1	1 case–control study	Serum and CSF	2−	Possible biomarker of DRE
IL-33	1 case–control study	Serum	2−	Possible biomarker of epilepsy.

## Data Availability

The data presented in this study are available upon request from the corresponding author.

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
