# Peer review of "A Systematic Review of the Predictive and Diagnostic Uses of Neuroinflammation Biomarkers for Epileptogenesis"

_ijms, 2024, doi:10.3390/ijms25126488_

Round 1
Reviewer 1 Report
Comments and Suggestions for Authors
The manuscript "Biomarkers of Neuroinflammation and Epileptogenesis: A Systematic Review" provides a valuable synthesis of current research on neuroinflammation and epilepsy. However, significant revisions are needed to address the weaknesses identified. I recommend a major revision to improve the manuscript's clarity, consistency, and depth of analysis. By addressing these issues, the authors can enhance the manuscript's impact and utility for researchers and clinicians.
1, While the manuscript includes a large number of studies, the quality of evidence is inconsistent. Many studies included are case-control studies, which have limitations compared to randomized controlled trials. For instance, the study by Kothur et al. (reference [36]) is rated as level 2-, indicating moderate to low quality of evidence.
2, Some sections of the manuscript are repetitive, particularly in the discussion of individual biomarkers. Streamlining the content would improve readability and reduce redundancy. The discussions on TNF-α and its receptors (sTNFr2) are repeated multiple times, with similar points made in different sections. For example, the role of TNF-α in neuroinflammation and its potential as a biomarker is discussed in both the results and discussion sections without adding new information.
3, The review primarily synthesizes existing literature without offering new insights or hypotheses. A more critical analysis and discussion of potential future research directions would add value.
4, Table 1 presents a summary of studies, but the level of detail varies. Some entries include comprehensive data, while others are sparser. Standardizing the level of detail would improve the clarity of the table.
5, Provide a more detailed discussion of the mechanisms by which neuroinflammation contributes to epileptogenesis. This could include more information on signaling pathways, molecular interactions, and potential therapeutic targets.
Author Response
RESPONSE TO REVIEWERS:
Reviewer 1:
The manuscript "Biomarkers of Neuroinflammation and Epileptogenesis: A Systematic Review" provides a valuable synthesis of current research on neuroinflammation and epilepsy. However, significant revisions are needed to address the weaknesses identified. I recommend a major revision to improve the manuscript's clarity, consistency, and depth of analysis. By addressing these issues, the authors can enhance the manuscript's impact and utility for researchers and clinicians.
A: Thank you for these kind words for our work.
1, While the manuscript includes a large number of studies, the quality of evidence is inconsistent. Many studies included are case-control studies, which have limitations compared to randomized controlled trials. For instance, the study by Kothur et al. (reference [36]) is rated as level 2-, indicating moderate to low quality of evidence.
A: Thanks for the comment. The level of the evidence was assessed using Scottish Intercollegiate Guidelines Network (SIGN) grading system.
In this system, the maximum degree of evidence (level 1) is only obtained by meta analyses, systematic reviews of randomised controlled trials (RCTs), RCTs with a very low or low risk of bias and systematic reviews.
Level 2 of evidence is reserved for case-control studies. This group includes most of the papers included in the systematic review. Among them, there are those classified as:
- 2++: High-quality case-control or cohort studies with a very low risk of confounding or bias and a high probability that the relationship is causal,
- 2+: Well-conducted case-control or cohort studies with a low risk of confounding or bias and a moderate probability that the relationship is causal
- 2-: Case-control or cohort studies with a high risk of confounding or bias and a significant risk that the relationship is not causal.
The distinction between risk of bias has been made based on criteria related to sample selection, the total number of patients, the number of healthy controls, and the clarity of description of procedures. This warning has been included in the legend of the table.
Finally, case series are classified as Level of evidence 3.
We have re-assigned levels to the different studies after a re-evaluation of them.
2, Some sections of the manuscript are repetitive, particularly in the discussion of individual biomarkers. Streamlining the content would improve readability and reduce redundancy. The discussions on TNF-α and its receptors (sTNFr2) are repeated multiple times, with similar points made in different sections. For example, the role of TNF-α in neuroinflammation and its potential as a biomarker is discussed in both the results and discussion sections without adding new information.
A: Thank you for the comment. Accordingly, we have removed all references to the role of individual molecules in neuroinflammation from the Results section, where only the results of the Systematic Review are included. All references to neuroinflammatory phenomena and the physiology of epileptogenesis have been relegated to the Discussion section.
3, The review primarily synthesizes existing literature without offering new insights or hypotheses. A more critical analysis and discussion of potential future research directions would add value.
A: Thank you for the comment. The discussion section has been revised and the development of the primary clinical consequences of the study of neuroinflammatory biomarkers in epilepsy, as well as treatment possibilities and research areas related to this topic, are covered in two new sections 3.3.1 and 3.3.2
4, Table 1 presents a summary of studies, but the level of detail varies. Some entries include comprehensive data, while others are sparser. Standardizing the level of detail would improve the clarity of the table.
A: I greatly appreciate your significant input. The table was, in fact, highly diverse. In an effort to improve the reader's comprehension and visual aid, we have reviewed and standardised all of the data. We now think that the text provides a much better explanation of the results and strength of evidence from every study.
5, Provide a more detailed discussion of the mechanisms by which neuroinflammation contributes to epileptogenesis. This could include more information on signaling pathways, molecular interactions, and potential therapeutic targets.
A: I think this is a very good comment and for this reason we have made two relevant changes:
1.- A figure summarizing the main neuroinflammatory mechanisms involved in epileptogenesis has been included.
2.- As previously mentioned, the Discussion section now specifically contemplates the potential targets derived from the more precise knowledge of inflammatory biomarkers.
All changes made to the original text are marked in red.
We hope that these modifications have improved the quality of the article and made it eligible for publication.
We sincerely appreciate all your work.
Fdo Pedro J. Serrano-Castro, MD; PhD on behalf of authors.

Reviewer 2 Report
Comments and Suggestions for Authors
Epilepsy is one of the most common neurological comorbidities and pro-epileptogenic context is a common condition for several neurological conditions. This is the reason why I consider this manuscript of potential interest.
Here are my concerns about the manuscript that I refer to the authors:
1) Title: I would suggest to modify it: the idea is Epileptogenesis diagnosis and prognosis improvement by identification of neuroinflammation biomarkers. You are not exploring separately neuroinflammation and epileptogenesis.
2) Abstract: Explain better the research strategy adding the database used and the combination of MesH terms during the research.
3) Introduction: Explain Seizure/Epilepsy and epileptogenesis terms. Association with different neurological conditions and different physiopathogenic mechanisms associated with proepileptic context beyond neuroinflammation.
Adding a figure summarizing these other physiopathogenic mechanisms and how neuroinflammation may alter the excitatory-inhibitory regulation of CNS could be of huge interest for the readers.
3) Methodology: Only Medline was used as a research Database (not WOS, Crossref...?); only original research were included (exclusion of narrative and systematic reviews and metaanalysis?); add extra information about research strategy (Mesh Term combinations, using AND or OR or other alternatives, directly in the text or in the flowchart); add information if specific programs for simultaneous checking of the utility of literature were used; explain a little bit better how abstracts were selected to final complete read... Move flowchart before table 1.
About specific molecule selection add a table/figure explaining easily their seleciton rationality, not only description of the design of the manuscript itself.
4) Discussion: Add a section about potential clinical practice implications (specific one) from diagnosis, prognostic and treatment prespective. In turn, add a section that related the neuroinflammation pathway with other pathogenic mechanisms related with epileptogenesis in different neurological conditions, and of course add a table about research gaps and future research ideas in the field.
Author Response
Reviewer 2:
Epilepsy is one of the most common neurological comorbidities and pro-epileptogenic context is a common condition for several neurological conditions. This is the reason why I consider this manuscript of potential interest.
Here are my concerns about the manuscript that I refer to the authors:
- Title: I would suggest to modify it: the idea is Epileptogenesis diagnosis and prognosis improvement by identification of neuroinflammation biomarkers. You are not exploring separately neuroinflammation and epileptogenesis.
R: We much appreciate your feedback, which will help to make our work's exposition more understandable. Actually, the goal of our article is to find neuroinflammation biomarkers associated with drug-resistant epilepsy and epileptogenesis. Regarding the reviewer's proposal, we offer the following title: A systematic review of the predictive and diagnostic uses of neuroinflammation biomarkers for epileptogenesis.
.
- Abstract: Explain better the research strategy adding the database used and the combination of MesH terms during the research.
R: Thank you for the comment. We have made the changes suggested in the Abstract. We have introduced this more explanatory paragraph:
The search strategy comprised the MESH and free terms of "Neuroinflammation" and selective searches for the following single biomarkers that had previously been found in narrative reviews: "High mobility group box 1 / HMGB1," "Toll-Like-Receptor 4 / TLR-4," "Interleukin-1 / IL-1," "Interleukin-6 / IL-6," "Transforming growth factor beta / TGF-β," and "Tumour necrosis factor-alpha / TNF-α." These queries were all combined with the MESH term "Epileptogenesis."
- Introduction: Explain Seizure/Epilepsy and epileptogenesis terms. Association with different neurological conditions and different physiopathogenic mechanisms associated with proepileptic context beyond neuroinflammation.
R: I appreciate your input. In that regard, we have updated the Introduction. Specifically, we have included two paragraphs to respond to this requirement:
“Epileptogenesis is the process by which a normal neural network becomes hyperexcita-ble and capable of causing seizures of epilepsy on its own[9]”
“The cross-functional nature of all these neuroinflammatory processes across various neurologic diseases may help to explain why the high incidence of epileptic seizures in patients with neurological pathologies as diverse as Multiple Sclerosis[23] or neurodegenerative disease[24]”.
And we've updated the references.
Adding a figure summarizing these other physiopathogenic mechanisms and how neuroinflammation may alter the excitatory-inhibitory regulation of CNS could be of huge interest for the readers.
R: Thank you for the comment. We have added a modified figure from a previous publication by our group that implicates the mechanisms of epileptogenesis.
- Methodology: Only Medline was used as a research Database (not WOS, Crossref...?); only original research were included (exclusion of narrative and systematic reviews and metaanalysis?);
A: Only references obtained from Medline have been included and we have preferred to eliminate all narrative reviews as well as expert opinions due to their low level of scientific evidence. However, systematic reviews and meta-analyses were included, so we thank the reviewers for their appreciation.
Indeed, 4 systematic reviews were identified and evaluated and finally excluded for the following reasons.
- Liew Y, Retinasamy T, Arulsamy A, Ali I, Jones NC, O'Brien TJ, Shaikh MF. Neuroinflammation: A Common Pathway in Alzheimer's Disease and Epilepsy. J Alzheimers Dis. 2023;94(s1):S253-S265. doi: 10.3233/JAD-230059. PMID: 37092226; PMCID: PMC10473147.
Exclusion Reason: This is a systematic review that focuses exclusively on epilepsy in patients with Alzheimer's disease.
- Costagliola G, Depietri G, Michev A, Riva A, Foiadelli T, Savasta S, Bonuccelli A, Peroni D, Consolini R, Marseglia GL, Orsini A, Striano P. Targeting Inflammatory Mediators in Epilepsy: A Systematic Review of Its Molecular Basis and Clinical Applications. Front Neurol. 2022 Mar 11;13:741244. doi: 10.3389/fneur.2022.741244. PMID: 35359659; PMCID: PMC8961811.
Exclusion reason: Systematic review focusing on treatment.
- Arulsamy A, Shaikh MF. Tumor Necrosis Factor-α, the Pathological Key to Post-Traumatic Epilepsy: A Comprehensive Systematic Review. ACS Chem Neurosci. 2020 Jul 1;11(13):1900-1908. doi: 10.1021/acschemneuro.0c00301. Epub 2020 Jun 16. PMID: 32479057
Exclusion reason: Articles that include basic research studies on tissues or animal models.
- Paudel YN, Shaikh MF, Chakraborti A, Kumari Y, Aledo-Serrano Á, Aleksovska K, Alvim MKM, Othman I. HMGB1: A Common Biomarker and Potential Target for TBI, Neuroinflammation, Epilepsy, and Cognitive Dysfunction. Front Neurosci. 2018 Sep 11;12:628. doi: 10.3389/fnins.2018.00628. PMID: 30271319; PMCID: PMC6142787.
Exclusion reason: Articles that analyzes general biomarkers of brain damage
We have modified the methods section, removing the reference to the exclusion of systematic reviews and specifying the exclusion criteria. Also the Flowchart has been modified.
add extra information about research strategy (Mesh Term combinations, using AND or OR or other alternatives, directly in the text or in the flowchart);
R: We have specified the search strategy by indicating the terms used in each search.
add information if specific programs for simultaneous checking of the utility of literature were used;
R: We have no used any specific program for checking the literature. This has been clarified in the text.
Explain a little bit better how abstracts were selected to final complete read...
R: Ok, we've included a brief description of how this selection of items was made.
Move flowchart before table 1.
R: Ok.
About specific molecule selection add a table/figure explaining easily their seleciton rationality, not only description of the design of the manuscript itself.
R: Figure 1, added to explain the rational hypothesis of the relationship of neuroinflammation biomarkers, identifies all biomarkers selected for selective searches.
- Discussion: Add a section about potential clinical practice implications (specific one) from diagnosis, prognostic and treatment prespective. In turn, add a section that related the neuroinflammation pathway with other pathogenic mechanisms related with epileptogenesis in different neurological conditions, and of course add a table about research gaps and future research ideas in the field.
A: I appreciate the suggestion; it's quite relevant. The development of the primary clinical consequences of the study of neuroinflammatory biomarkers in epilepsy, as well as treatment possibilities and research areas related to this topic, are covered in two new sections 3.1 and 3.2
All changes made to the original text are marked in red.
We hope that these modifications have improved the quality of the article and made it eligible for publication.
We sincerely appreciate all your work.
Fdo Pedro J. Serrano-Castro, MD; PhD on behalf of authors.

Round 2
Reviewer 2 Report
Comments and Suggestions for Authors
I really would like to thank the authors because they have properly addressed all my concerns and have significantly improved the manuscript quality, and as I previously refer I consider that the work could be of potential interest both for clinical (mainly after having done a more clinical approach in different sections in comparison with the original manuscript version) and translational researchers studying epileptogenosis, not only in complex epilepsy.